REVIEW-SYMPOSIUM

# Neurobiology of mitochondrial dynamics in sleep

Raffaele Sarnataro ⬤

*Centre for Neural Circuits and Behaviour, University of Oxford, Oxford, UK*

Handling Editors: Laura Bennet & Diana Martinez

The peer review history is available in the Supporting Information section of this article (https://doi.org/10.1113/JP288054#support-information-section).

**Abstract figure legend** In neurons, variations in sleep history are accompanied by alterations in mitochondrial dynamics, including changes in size, fission and fusion events, the emergence of extra-large and 'hourglass'-shaped mitochondria, mitophagy, and modifications in intracristal space volume and in the number of contact sites between mitochondria and endoplasmic reticulum.

**Abstract** The cycling of sleep and wakefulness reshapes neuronal activity, gene expression, and cellular metabolism of the brain. Such reshuffling of brain metabolism implicates key mediation by mitochondria. Mitochondrial dynamics enable organelles to adapt their morphofunction to changing metabolic demands, and experimental evidence increasingly links these processes to sleep-wake regulation. Across species, sleep loss perturbs mitochondrial gene expression, increases oxidative stress, and disrupts organelle structure, particularly in energy-demanding brain regions. In *Drosophila*, sleep-control neurons projecting to the dorsal fan-shaped body (dFBNs) exhibit a homeostatic feedback mechanism coupling mitochondrial activity to behavioural state. As sleep pressure elevates, dopaminergic inhibition reduces dFBN excitability

**Dr Raffaele Sarnataro** is a Fulford Junior Research Fellow at Somerville College, Oxford and a Postdoctoral Fellow at the University of Oxford, in the Miesenböck group. His research focuses on understanding how the need for sleep is controlled and encoded in the brain by studying the bioenergetic molecular machinery and network dynamics of sleep-control neurons. He was educated in molecular and cell biology at Scuola Normale Superiore and University of Pisa, Italy, and in neuroscience at the University of Oxford, UK, where he obtained his MSc and DPhil. Before Oxford, he worked at Harvard Medical School and Scuola Normale Superiore.

The Journal of Physiology

and ATP consumption, triggering mitochondrial fission and accumulation of reactive oxygen species (ROS) that biochemically prime the neurons for subsequent sleep induction. Upon relief of inhibition during recovery sleep, dFBNs elevate their activity, consume ATP, and undergo mitochondrial fusion to restore energy balance. Artificial modulation of mitochondrial morphology and ATP production in these neurons bidirectionally alters sleep. dFBNs' elevated OxPhos expression and mitochondrial turnover render them sensitive to metabolic shifts and capable of encoding internal states. While dFBNs remain the only known neurons where mitochondrial dynamics are coupled to sleep behaviour, other populations, like mammalian cortical neurons or fly Kenyon cells, also display mitochondrial changes after sleep loss. Sleep, like other state-dependent behaviours including hunger and memory, imposes shifting energetic demands on specific neuronal populations. Mitochondrial dynamics may thus provide a conserved, cell-autonomous mechanism for tuning neural excitability and sleep pressure, enabling brain-wide coordination of metabolic and behavioural homeostasis.

(Received 26 May 2025; accepted after revision 4 August 2025; first published online 22 August 2025)

**Corresponding author** R. Sarnataro: Centre for Neural Circuits and Behaviour, University of Oxford, Tinsley Building, Mansfield Road, Oxford, OX1 3SR, UK.    Email: raffaele.sarnataro@cncb.ox.ac.uk

## Introduction

Harboured by cells, mitochondria retain features reminiscent of their bacterial ancestors: they can divide and fuse, move and concentrate where it is most advantageous and be phagocytised to maintain population fitness (Friedman & Nunnari, 2014). While the machineries enabling these changes have diverged from or displaced the bacterial ones, mitochondria have remained highly dynamic organelles (Kiefel et al., 2006; Lewis & Lewis, 1914). In fact, the processes of division (or fission, or fragmentation), fusion, transport, and cellular phagocytosis of mitochondria (or mitophagy) are collectively known as 'mitochondrial dynamics' (Bulthuis et al., 2019; Cagalinec et al., 2013; Tábara et al., 2025).

Mitochondria serve as the central cellular energy and metabolite production hubs, and since nervous systems are among the most energetically demanding tissues (Attwell & Laughlin, 2001), neurons orchestrate mitochondrial dynamics throughout their complex and extended arborisations to sustain the metabolic requirements of the neural computations underlying behaviours (López-Doménech & Kittler, 2023; Mann et al., 2021; Seager et al., 2020).

Here we will review how mitochondrial dynamics are involved in a behaviour that may be as ancestral as the first nervous systems: sleep (Allada & Siegel, 2008; Nath et al., 2017).

To understand how a subcellular process and a behaviour might be mechanistically linked, we will first examine the known mechanisms by which mitochondrial dynamics are coupled to neuronal activity. We will then discuss how neuronal bioenergetics is shaped by sleep and inspect the experimental evidence that implicates mitochondrial dynamics in sleep behaviour.

## Mitochondrial dynamics are coupled to neuronal activity

Neurons are metabolically demanding cells, expending the greatest proportion of energy by spiking and synaptic transmission, whose patterns and plasticity, respectively, are the bases for neural computations. Mitochondrial dynamics can adjust to the energetic requirements underlying the appropriate neuronal functions (Burté et al., 2015; López-Doménech & Kittler, 2023; Misgeld & Schwarz, 2017). Glaring evidence of this key role is the observation that neurons are particularly vulnerable to defects in the machinery enacting mitochondrial dynamics, often resulting in neurodegenerative disorders (Alexander et al., 2000; Cho et al., 2009; Niemann et al., 2005; Pickrell & Youle, 2015; Song et al., 2011; Wang et al., 2009; Waterham et al., 2007; Züchner et al., 2004).

Neurons commonly deploy various mitochondrial morphologies through their complex arborisations and functional compartmentalisations (Burté et al., 2015; López-Doménech & Kittler, 2023; Misgeld & Schwarz, 2017): for example, in cortical and CA1 hippocampal pyramidal neurons, while dendritic mitochondria display long and tubular shapes, axonal mitochondria are more fragmented (Faitg et al., 2021; Lewis et al., 2018). Even within the dendritic arbour, pyramidal neurons in CA1 of the hippocampus display an array of mitochondrial morphologies that range from highly fused and elongated shapes in the apical tuft to more fragmented mitochondria in dendritic compartments proximal to the soma, in functional correspondence to different sets of synaptic inputs (Lee et al., 2022; Virga et al., 2024).

While these variations could merely reflect dissimilarities in anatomical and cellular spatial constraints of different subcellular compartments,

mitochondria do dynamically change their shapes and local density to satisfy varying energetic demands and trigger neuronal signalling cascades. How do mitochondrial dynamics shape neuronal activity? Primarily via the modulation of local ATP concentration, the generation of reactive oxygen species (ROS), and calcium handling. Moreover, the metabolism and recycling of mitochondrially derived metabolites (importantly, neurotransmitters glutamate and GABA) may represent an underexplored additional mechanistic route (Andersen, 2025).

The consumption of energy, primarily used for restoring ion gradients in neurons, represents a fundamental constraint on how the brain processes information. Elevated levels of ATP are required to sustain energy demand across different firing rates and activity patterns (Harris et al., 2012; Rangaraju et al., 2014; Yi & Grill, 2019), and can themselves change neuronal excitability via the closure of ATP-sensitive potassium channels ($K_{ATP}$). In the sleep- and energy-control neurons synthesising hypocretin/orexin in the lateral hypothalamus, intracellular ATP levels rise from about 4–5 mM in rested mice to 5–6 mM in sleep-deprived animals and shift a larger proportion of $K_{ATP}$ channels towards their closed state, thus depolarising their membrane potential (Liu et al., 2011). In the sleep-control neurons of the fly projecting to the dorsal fan-shaped body (dFBNs), ATP levels rise by about 20% after sleep deprivation (Sarnataro et al., 2025), anticipating higher sleep-inducing neuronal activity (Donlea et al., 2014; Hasenhuetl et al., 2024).

While re-establishing ion gradients after spiking requires ATP, a plethora of ROS-sensing components alter neuronal membrane excitability itself (Doser & Hoerndli, 2021), ranging from voltage-gated potassium channels (Kempf et al., 2019; Tipparaju et al., 2005, 2008; Weng et al., 2006) to $GABA_A$ receptors (Amato et al., 1999; Accardi et al., 2014; Beltrán González et al., 2020), to the modulation of activity-dependent transport of AMPA receptors (Doser et al., 2020).

Neuronal mitochondria help buffer calcium by taking up excess $Ca^{2+}$ during neuronal activity, thus shaping spatiotemporal features of cytosolic $Ca^{2+}$ waves. A fragmented mitochondrial network hampers the spread of intramitochondrial $Ca^{2+}$ waves in cultured cells by leaving a fraction of individual mitochondria without substantial calcium elevation (Szabadkai et al., 2004), resulting in a slower buffering of the $Ca^{2+}$ entering across the plasma membrane (Frieden et al., 2004). This tuning of $Ca^{2+}$ buffering by the mitochondria clamps the cytosolic $Ca^{2+}$ concentration, affecting, among other processes, synaptic release (Ashrafi et al., 2020; Billups & Forsythe, 2002), calcium-activated slow afterhyperpolarisation (Groten & MacVicar, 2022), and ATP production (McCormack et al., 1990).

Since ATP synthesis, ROS production, and calcium homeostasis are interconnected processes and each of them is mutually coupled to mitochondrial dynamics (Mishra & Chan, 2016; Szabadkai et al., 2006), it is plausible that combinations of those processes contribute to modulating neuronal activity. Therefore, we will examine how each class of mitochondrial dynamics affects neuronal activity.

**Mitochondrial fission and fusion.** Fusion is the merging of two mitochondria into a single, elongated organelle; this process allows mixing of the mitochondrial contents and promotes metabolic efficiency (Bulthuis et al., 2019; Tábara et al., 2025). Fusion is mediated by mitofusins (Mfn-1/2 in mammals, Marf in flies) on the outer and Opa1 on the inner mitochondrial membrane (Bulthuis et al., 2019; Tábara et al., 2025).

In contrast, fission splits one mitochondrion into two smaller units, supporting organelle distribution across the cell and facilitating the isolation of damaged mitochondria. Fission is enacted by Drp1, a cytosolic GTPase recruited to the mitochondrial surface by receptors like Fis1 (Bulthuis et al., 2019; Cagalinec et al., 2013; Tábara et al., 2025).

These dynamics are modulated by post-translational modifications of their core regulatory proteins, by the intracellular redox state (Willems et al., 2015), and by mitochondrial lipids (Kameoka et al., 2018). Interestingly, daily oscillations in Drp1 and Opa1 activity, and changes in mitochondrial morphology and handling of ROS follow a circadian rhythm, suggesting a circadian control of the balance between fission and fusion (Sardon Puig et al., 2018; Schmitt et al., 2018).

Such balance is coupled to neuronal activity: inhibiting neuronal activity using tetrodotoxin raises the ratio of mitochondrial fusion to fission in dendrites, whereas depolarising neurons with potassium chloride leads to opposite effects (Li et al., 2004). Mitochondrial fission is essential for segregating damaged mitochondria, thereby preventing the accumulation of dysfunctional components that could impair ATP synthesis and contribute to cellular stress (Youle & Van Der Bliek, 2012).

This process is of particular relevance in synapses, which are densely populated by mitochondria (Fu et al., 2017). An appropriate fusion/fission balance sustains synaptic formation and maintenance during development (Ishihara et al., 2009; Li et al., 2004; Wakabayashi et al., 2009), and is required for synaptic plasticity in mature circuits (Divakaruni et al., 2018; Fu et al., 2017; Verstreken et al., 2005). In hippocampal neurons, long-term potentiation-inducing protocols elevate $Ca^{2+}$, triggering mitochondrial fission within minutes, which elevates mitochondrial matrix $Ca^{2+}$; conversely, halting fission impairs mitochondrial matrix

$Ca^{2+}$ elevations and structural and electrophysiological long-term potentiation (Divakaruni et al., 2018).

ATP levels are also regulated by the fission/fusion balance. The most widely accepted framework suggests that mitochondrial fusion increases the efficiency of oxidative phosphorylation (OxPhos) by reducing proton leak and elevating mitochondrial membrane potential (Hoitzing et al., 2015; Picard et al., 2013). This results in higher ATP production, which is particularly beneficial during periods of increased energy demand (Hoitzing et al., 2015). The relationship between OxPhos and mitochondrial fusion seems bidirectional: in mammalian isolated mitochondria and cell cultures, promoting OxPhos metabolism stimulates mitochondrial fusion by enhancing proteolytic cleavage of Opa1, and thus its activity (Mishra et al., 2014).

In mouse brain and fibroblasts, along the circadian day, the interconnectivity of the mitochondrial network correlates directly with ATP levels and OxPhos activity, and inversely with Drp1 activity (Schmitt et al., 2018); analogously, in fly dFBNs, overexpression of Drp1 lowers ATP levels and fissures mitochondria in their dendritic processes (Sarnataro et al., 2025). Agouti-related peptide and neuropeptide-Y (Agrp/NPY) neurons in the arcuate nucleus of the hypothalamus of mammals are more hyperpolarised if mitofusin-1 or -2 is knocked out, compared to wild-type animals (Dietrich et al., 2013). This difference in membrane voltage can be abolished by dialysing high intracellular concentrations of ATP (Dietrich et al., 2013). Since $K_{ATP}$ channels operate in Agrp/NPY neurons (Belgardt et al., 2009; Spanswick et al., 1997), the observed neuronal hyperpolarisation suggests that the knockout of mitofusins reduces ATP levels. [Correction made on 29th September 2025, after first online publication: The phrase has been corrected to read "analogously, in fly dFBNs, overexpression"]

However, when glycolysis is pharmacologically blocked, hippocampal neurons from mice lacking Drp1 are unable to sustain, in axons but not in cell bodies, the ATP levels required by electrical stimulation to the point of impairing synaptic vesicle recycling (Shields et al., 2015). This indicates that a physiological fusion/fission balance is necessary for generating mitochondrial-derived ATP in the relevant cell compartments.

This body of evidence suggests that bioenergetics rules seem to exist (Hoitzing et al., 2015; Liesa & Shirihai, 2013): more activity promotes more fusion to fuel higher ATP production, and conversely for fission. However, the extent of the coupling between ATP levels and the shape of mitochondria depends on the functional specificities of the neuronal compartment: intuitively, their energy requirements.

**Mitochondrial transport.** Mitochondria are actively transported along microtubules to energetically demanding sites, such as synapses, nodes of Ranvier, growth cones, axon terminals, and dendritic spines (Fukumitsu et al., 2015; Hollenbeck & Saxton, 2005; Sheng & Cai, 2012), to ensure an abundance of mitochondria where ATP demand and calcium buffering requirements are elevated (Cagalinec et al., 2013).

A high calcium concentration, possibly resulting from high local synaptic activity, allows release of mitochondria from the machinery that anchors them to microtubules; this hinders their mobility, thus accumulating mitochondria and affecting short-term facilitation (Chen & Sheng, 2013; MacAskill et al., 2009; Wang & Schwarz, 2009). Not only synaptic activity, but also cellular nutrient levels directly influence organelle trafficking: under high-glucose conditions, glycosylation of the mitochondrial transport protein Milton immobilises mitochondria. This concentrates mitochondria in nutrient-rich regions, and thereby optimises ATP synthesis efficiency (Pekkurnaz et al., 2014). Anterograde transport of mitochondria into axons is also promoted in conditions of ATP depletion or hypoxia (Li et al., 2009; Mironov, 2007; Tao et al., 2014).

**Mitophagy.** Mitophagy is the selective autophagic degradation of dysfunctional mitochondria, by which damaged organelles are enclosed within autophagosomes for lysosomal degradation (Tábara et al., 2025). Neurons employ mitophagy as a quality control mechanism to maintain mitochondrial integrity and support energy homeostasis. Excessive mitochondrial damage triggers fission, followed by selective degradation via mitophagy (Palikaras et al., 2018). While the PINK1/Parkin pathway is the most studied, alternative mechanisms involving cardiolipins and receptors such as BNIP3, NIX, and FUNDC1 also contribute to mitochondrial clearance, especially under stress or the presence of developmental cues (Palikaras et al., 2018).

In neurons, mitophagy is spatially regulated: it predominantly occurs in the somatodendritic compartment where mature lysosomes reside, with retrograde transport bringing damaged mitochondria to the soma for degradation (Cai et al., 2012; McWilliams et al., 2016). However, under acute mitochondrial stress, local mitophagy in axons can be initiated, involving the recruitment of autophagic machinery directly to damaged organelles (Ashrafi et al., 2014). Neurons also engage in non-cell-autonomous mitophagy: retinal ganglion cells, for example, offload mitochondria to astrocytes at the optic nerve head (Davis et al., 2014). Functionally, mitophagy facilitates metabolic rewiring, such as shifts toward glycolysis during differentiation (Esteban-Martínez et al., 2017).

## Neuronal bioenergetics of sleep

Mitochondrial dynamics equip the nervous system with an arsenal to meet changing energy demands. This flexibility supports the neural computations underlying behavioural states.

Among those, changes in vigilance states are behaviourally the most dramatic transition that any animal experiences daily and remarkably reshape the electrical and metabolic landscape of brains. Neuronal mitochondria require from a few tens of seconds to several minutes to complete the fission and fusion processes (Cagalinec et al., 2013), timescales which are compatible with sleep state transitions (Saper et al., 2010). Through such transitions, the electrical activity of a sleeping and an awake brain, may it be of a mammal, reptile, fish, or fly (Campbell & Tobler, 1984; Leung et al., 2019; Yap et al., 2017), differs remarkably. In mammals, for example, during NREM sleep, large-scale cortical activity is marked by slow waves (low-frequency, high-amplitude EEG signals) reflecting synchronised cycles of depolarised 'up' states and hyperpolarised 'down' states in certain cortical neurons. In contrast, wakefulness drives these neurons into a more depolarised, desynchronised firing pattern, resulting in high-frequency, low-amplitude activity (Saper et al., 2010; Steriade et al., 1993).

Slow wave activity power at the onset of sleep correlates with the slope of cortical evoked field potentials, which reflect enhanced synaptic strength, after a period of wakefulness (Tononi & Cirelli, 2014; Vyazovskiy, Cirelli, Pfister-Genskow, et al., 2008). This and further experimental evidence support one of the most influential hypotheses on the main function of sleep, the 'synaptic homeostasis hypothesis' (SHY) (Tononi & Cirelli, 2003, 2014). SHY maintains that a solid disconnection from the external world, sleep, is necessary for renormalising the net synaptic weights, following the experiences of the awake brain (De Vivo et al., 2017; Tononi & Cirelli, 2014; Vyazovskiy, Cirelli, Pfister-Genskow, et al., 2008). Since synaptic activity consumes most of the neurons' energy (Harris et al., 2012; Rangaraju et al., 2014), and that about 95% of cerebral energy metabolism is fuelled by mitochondrial OxPhos (Sokoloff, 1984), the sleep-wake transition indeed alters neuronal aerobic energy allocation (Scharf et al., 2008).

Consistent with this view that synaptic downscaling during sleep contributes to the metabolic homeostasis of the brain, supporting its optimal function and capacity for plasticity (Tononi & Cirelli, 2014), glucose uptake is high during wake and low during sleep in human cerebral blood flow (Boyle et al., 1994) and rodent cortex (Ramm & Frost, 1983; Vyazovskiy, Cirelli, Tononi, et al., 2008). Correspondingly, the transition from NREM or REM sleep to wake elevates ATP levels in excitatory neurons of the mouse cortex (Natsubori et al., 2020).

Part of the surges in local energy demand provoked by cognition and synaptic activity proper of awake brains are met also with the support of glial-derived lactate (Magistretti & Allaman, 2018). Astrocytes convert glucose to lactate via glycolysis, then export lactate to neurons, in a process called astrocyte-neuron lactate shuttle (ANLS) (Magistretti & Allaman, 2018). Neurons take up this lactate, oxidising it in mitochondria to efficiently fuel synaptic activity and memory processes (Pellerin & Magistretti, 1994). For example, learning leads to a significant increase in extracellular lactate levels in rat hippocampus: there, ANLS is necessary, in fact, for the synaptic strengthening that supports long-term memory formation (Suzuki et al., 2011). Indeed, transitions across vigilant states are paralleled by variation in levels of lactate: in mammalian cortex, lactate rapidly and sustainedly increases upon waking and during REM sleep, while it consistently declines during NREM (Bellesi et al., 2018; Lundgaard et al., 2017; Naylor et al., 2012).

The different energy currency levels also directly modulate the excitability of key sleep-control brain regions, from humans to flies (Allebrandt et al., 2013), such as hypocretin/orexin neurons of the lateral hypothalamus (Liu et al., 2011) or cortex (Constantino et al., 2025), by the ATP-mediated closure of $K_{ATP}$ channels.

This evidence suggests sleep maintains the homeostasis of brain metabolism: indeed, loss of sleep alters mitochondrial and metabolic gene expression levels in the brain: 12S rRNA, subunits of NADH dehydrogenase and $F_1$-ATPase, subunits and activity of cytochrome c oxidase (Cirelli & Tononi, 1998; Nikonova et al., 2005, 2010; Shaw et al., 2000), uncoupling protein 2, phosphorylated AMPK, transcription factors regulating mitochondrial function and biogenesis (Nikonova et al., 2010), the mitochondrial chaperon HSPA9, the outer mitochondrial membrane Voltage-Dependent Anion Channel 1, and other genes involved in respiratory chain activity and mitochondrial ribosomal subunit assembly (Aboufares El Alaoui et al., 2023). These warped processes are accompanied by, and might underlie (Chintaluri & Vogels, 2023; Hill et al., 2018; Ikeda et al., 2005; Kempf et al., 2019; Sarnataro et al., 2025), the generation of ROS (Kempf et al., 2019; Rodrigues et al., 2018; Silva et al., 2004; Vaccaro et al., 2020; Villafuerte et al., 2015), or the decrease of antioxidant defences (D'Almeida et al., 1998; Ramanathan et al., 2002), observed during extended sleep deprivation in various animals' brain regions. Lack of sleep manifests also in other metabolic pathways, such as unfolded protein response, glycogen storage and adenosine accumulation (Melhuish Beaupre et al., 2022; Scharf et al., 2008), which collectively imply that wakefulness imposes a significant metabolic challenge to the brain, whereas sleep facilitates the restoration of cerebral energy homeostasis, a view known as 'the energy hypothesis of sleep' (Benington & Craig Heller, 1995; Scharf et al., 2008).

## Sleep and mitochondrial dynamics

Given this evidence, it is therefore plausible that rearrangements in mitochondria accompany, or even underpin, transitions across vigilance states: this is corroborated by a series of ultrastructural and cellular observations correlated to sleep history (Fig. 1).

The prefrontal cortex strengthens its synapses upon sleep loss, shrinks them during sleep and is a main source of EEG slow waves (Nir et al., 2011; Sawada et al., 2024). Sleep deprivation protocols, including acute sleep deprivation and chronic sleep restriction, provoke rearrangements in the ultrastructure of mitochondria of the pre-frontal cortex of adolescent mice (De Vivo et al., 2016): bigger sized mitochondria, occupying a larger fraction of cytoplasm, a larger proportion of 'hourglass'-shaped, and the appearance of extra-large mitochondria significantly accompany chronic sleep restriction (consisting in an average decrease of about 70% in total sleep time compared to unperturbed baseline, over 4 days), and show comparable trends upon acute sleep deprivation (De Vivo et al., 2016) (Fig. 1). These changes probably echo the elevated energetic demand imposed by extended wakefulness to cortex (DiNuzzo & Nedergaard, 2017; Vyazovskiy et al., 2009), of which a visible testimony is the 50% elevation, in cortex of sleep-deprived mice, of contact sites between mitochondria and endoplasmic reticulum (MERCS) (Aboufares El Alaoui et al., 2023) (Fig. 1): across those, calcium and lipids are shuffled for boosting ATP production, enhancing phospholipids production rates, and even the production of sleep-controlling neurotransmitter vesicles (Valadas et al., 2018). Another marker of awake brains, high lactate, directly promotes the increase in mitochondrial biomass in mammalian cortex and hippocampus (Akter, Hasan, et al., 2023; Akter, Ma, et al., 2023).

If some of these changes might be compensatory mechanisms for the enhanced metabolic needs of extended wakefulness, prolonged (72 h) REM sleep deprivation cascades to additional and more dramatic ultrastructural mitochondrial alterations in the forebrain (Fig. 1). Notably, reductions in intracristal space volume are observed in the hippocampus and neocortex of sleep-deprived rats (Lu et al., 2021), accompanied, in the hippocampus, by the expression of mitophagy-related proteins (Dai et al., 2021), and even by pro-apoptotic signals such as elevated levels of the mitochondrial protein Bax and cytosolic release of cytochrome c (Yang et al., 2008) (Fig. 1). Interestingly, while mitochondrial cristae remodel during apoptosis (Frezza et al., 2006), they have also been shown following non-cell-death stimuli to adapt to metabolic demand (Patten et al., 2014), even in forebrain neurons after altering the nutritional state (Gómez-Valadés et al., 2021): this points to a metabolic adaptive effect preceding apoptosis.

These findings suggest the possibility that mitochondrial changes correlate with the sleep state-related energetic demands of the neural population. Second-order excitatory olfactory projection neurons (PNs) of *Drosophila*, for example, do not alter sleep if artificially activated (Hsu et al., 2021), nor change their intracellular ATP levels upon sleep loss (Sarnataro et al., 2025); they do not bear transcriptional (Dopp et al., 2024; Sarnataro et al., 2025) or structural (Sarnataro et al.,

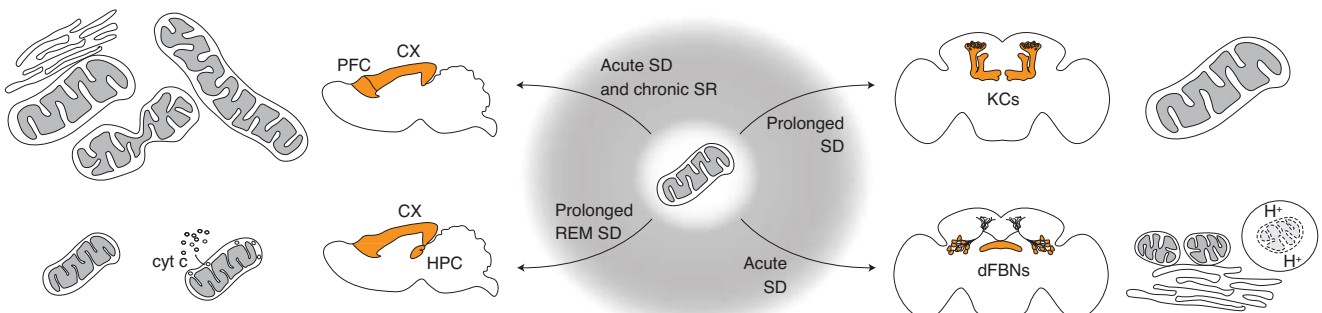

**Figure 1. Sleep loss alters mitochondrial dynamics**

Lack of sleep (shaded grey area) provokes changes in mitochondrial dynamics in the brain of a rodent (left) and fly (right). Top, left: in the mouse cortex (CX), acute sleep deprivation (SD) elevates the number of contact sites between mitochondria and endoplasmic reticulum (MERCS) (Aboufares El Alaoui et al., 2023); in murine pre-frontal cortex (PFC), bigger-sized mitochondria and the appearance of extra-large and more 'hourglass'-shaped mitochondria are significantly detected after chronic sleep restriction (SR, an average decrease of about 70% in total sleep time compared to unperturbed baseline, over 4 days), with comparable trends upon acute SD (De Vivo et al., 2016). Bottom, left: Prolonged (72 h) deprivation of REM sleep reduces intracristal space volume in the hippocampus (HPC) and neocortex (CX) of rats (Lu et al., 2021); in HPC, this is accompanied by the release of cytochrome c into cytosol (Yang et al., 2008). Top, right: prolonged (35 h) SD induces a trend towards an increase in the percentage of cell volume occupied by mitochondria in fly Kenyon cells (KCs) (Flores et al., 2022). Bottom, right: acute SD fissures mitochondria, increases the number of MERCS, and elevates levels of mitophagy in neurons projecting to the dorsal fan-shaped body (dFBNs) of the fly (Sarnataro et al., 2025).

2025) traces of mitochondrial rearrangements following sleep deprivation, and the artificial manipulation of mitochondrial dynamics core machinery does not alter sleep (Sarnataro et al., 2025). Interestingly, Kenyon cells (KCs), just one synapse downstream from PNs in the fly olfactory processing pathway, vary their activity depending on the sleep state (Bushey et al., 2015), and display a significant transcriptional response to sleep loss, despite not showing a metabolic one (Dopp et al., 2024; Sarnataro et al., 2025), and a trend towards an increase in the percentage of cell volume occupied by mitochondria only after prolonged (35 h) sleep deprivation (Flores et al., 2022) (Fig. 1).

However, when the mitochondrial dynamics machinery is altered in KCs, no behavioural effects are detected (Sarnataro et al., 2025). Since the KC population comprises sleep- and wake-inducing neurons (Sitaraman et al., 2015), and given that sleep loss affects synaptic strength differently in different KC subtypes (Weiss & Donlea, 2021), it is possible that manipulation of the dynamics machinery in the whole KC population may produce a mixture of opposing behavioural drives without a distinct behavioural readout, similarly to their variegated global transcriptional response to sleep loss (Sarnataro et al., 2025). Cell-type-specific manipulations and readouts might disentangle and reveal, if any, relevant mitochondrial dynamics in KC subpopulations.

**The case of dorsal fan-shaped body neurons of the fly brain.** The coupling between the different bioenergetic needs imposed by sleep and wakefulness, and their implementation via mitochondrial dynamics, is tighter and evident in a population of just over two dozen sleep-control neurons of the fly brain. A set of neurons projecting to the dorsal fan-shaped body enhances their activity as sleep pressure accrues (Donlea et al., 2014; Hasenhuetl et al., 2024; Raccuglia et al., 2025), and if artificially activated or inhibited, can impose or relieve sleep and sleep rebound on demand (Donlea et al., 2011; Hasenhuetl et al., 2024; Hsu et al., 2025; Jones et al., 2025; Keleş et al., 2025; Pimentel et al., 2016). [Correction made on 29th September 2025, after first online publication: The phrase has been corrected to read "A set of neurons projecting to the dorsal fan-shaped body enhances"]

In dFBNs, the interplay between mitochondrial structure, cellular metabolic state, and sleep regulation is coordinated by a homeostatic feedback mechanism that arises from the equilibrium between production and demand of cellular ATP (Sarnataro et al., 2025) (Fig. 2).

During daytime, dFBNs activity is low (Raccuglia et al., 2025), similar to enforced wakefulness, when it is abated by dopaminergic inputs (Hasenhuetl et al., 2024; Pimentel et al., 2016). The low ATP demand during such vigilance states (Sarnataro et al., 2025), when caloric intake is high but the neurons' electrical activity is reduced, causes mitochondrial electron leak and elevated ROS production (Kempf et al., 2019; Liesa & Shirihai, 2013; Murphy, 2009) (Fig. 2). dFBNs respond through reduction of electron flow, by fissuring their mitochondria, shifting them towards a 'less efficient' bioenergetic state that accompanies the depression of

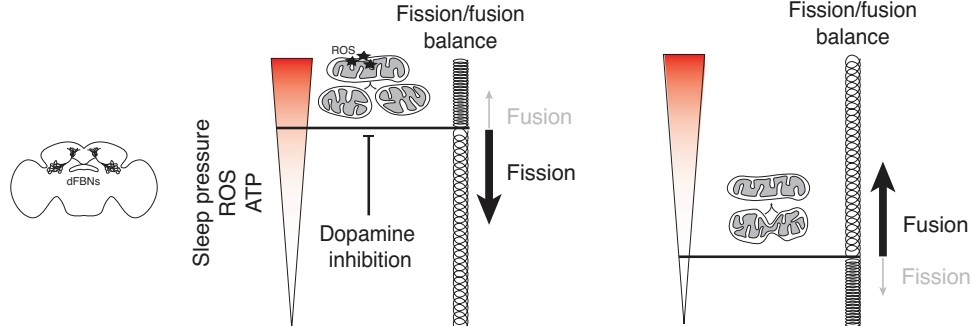

**Figure 2. Mitochondrial dynamics and sleep in dFBNs**
High intracellular levels of ATP and ROS in dFBNs enhance their excitability, thus elevating sleep pressure, through energy currency availability and modulation of a redox-sensitive potassium channel (Kempf et al., 2019; Sarnataro et al., 2025). Left: when arousing dopaminergic stimuli inhibit dFBNs, for example during enforced wakefulness, dFBNs exhibit low activity and ATP consumption, leading to mitochondrial electron leak and elevated ROS (Hasenhuetl et al., 2024; Kempf et al., 2019; Pimentel et al., 2016; Sarnataro et al., 2025). In response to the elevated intracellular oxidative state, dFBNs reduce electron flow via mitochondrial fission to restore metabolic cellular homeostasis. They also trigger restorative mechanisms, including mitophagy, increased MERCS, and mitochondrial gene expression (Sarnataro et al., 2025). Right: once the arousing stimuli terminate, the heightened excitability drives higher neuronal activity, which imposes recovery sleep and elevates ATP consumption: its demand is met by mitochondrial fusion and mass increase, restoring bioenergetic homeostasis (Sarnataro et al., 2025). Manipulating mitochondrial dynamics in dFBNs directly modulates sleep, mimicking homeostatic responses (Sarnataro et al., 2025). Artificially inducing fission in the absence of external inhibitory stimuli lowers excitability, ATP levels and sleep. Inducing fusion has the opposite effects (Sarnataro et al., 2025).

their efferent synapses (Hasenhuetl et al., 2024), and by deploying several restorative measures: (i) promoting mitophagy to replace mitochondria damaged by ROS, (ii) increasing the number of MERCS, possibly to shuffle calcium to sustain mitochondrial activity and replenish peroxidised lipids (Rorsman et al., 2025), and (iii) by transcribing more mitochondrial genes, priming organelle biogenesis for recovery sleep (Sarnataro et al., 2025) (Figs 1 and 2).

Inducing fission artificially mimics such homeostatic response: fission reduces ATP levels, spontaneous and rebound sleep, and attenuates excitability; conversely, promoting fusion has opposite effects, i.e. an increase of spontaneous and rebound sleep, and enhancement of excitability (Sarnataro et al., 2025).

High sleep pressure is thus represented cell-autonomously in dFBNs by an elevation of both ATP (Sarnataro et al., 2025), whose abundance can sustain higher activity (Hasenhuetl et al., 2024) and different firing patterns (Chintaluri & Vogels, 2023), and ROS (Kempf et al., 2019).

The elevated intracellular oxidative state promotes the conversion of the Shaker channel's $K_V\beta$ subunit, Hyperkinetic, into its $NADP^+$-bound form (Kempf et al., 2019). The oxidised cofactor slows the inactivation of A-type potassium currents, leading to increased action potential firing rates and the promotion of sleep (Kempf et al., 2019).

The system resets following recovery sleep, when dFBNs are relieved of the arousing inhibitory dopaminergic inputs. This permits spiking of dFBNs with biochemically elevated excitability, which in turn induces sleep (Donlea et al., 2011, 2014; Hasenhuetl et al., 2024; Jones et al., 2025), boosting the consumption of ATP (Sarnataro et al., 2025), whose higher demand is met by the fusion of the mitochondrial network and the elevation of mitochondrial mass (Sarnataro et al., 2025) (Fig. 2). In fact, when the protonmotive force that generates ATP in the mitochondria is artificially elevated by a proton pump that uses photons instead of matrix electrons, sleep is instigated (Sarnataro et al., 2025). Symmetrically, when uncoupling proteins, dissipating protonmotive force, are expressed in dFBNs, sleep is reduced (Sarnataro et al., 2025).

dFBN activity is modulated not only by the described cell-intrinsic account of sleep need represented by the cell's excitability but also by synaptic inputs. Interestingly, many of these network connections are recurrent: dFBNs directly inhibit themselves (Hasenhuetl et al., 2024), are in a feedback loop with the abovementioned inhibitory dopaminergic neurons (Hasenhuetl et al., 2024; Hulse et al., 2021), with ring neurons of the central complex (Donlea et al., 2018; Hasenhuetl et al., 2024; Liu et al., 2016), and with octopaminergic (Hulse et al., 2021; Ni et al., 2019) and fan-shaped body columnar (Hulse et al., 2021) neurons. Due to its recurrent architecture, this circuitry can exhibit emergent dynamics which may underlie flexible control of sleep–wake behaviour, including oscillations (Hasenhuetl et al., 2024) or a synaptic depression (Hasenhuetl et al., 2024; Sarnataro et al., 2025) that is anticyclical to most of the sleep-deprived brain (Gilestro et al., 2009). Thus, the feedback mechanism that in dFBNs tracks sleep need and adjusts their activity accordingly arises from the interaction of their highly recurrent circuitry with a particularly sensitive cell-autonomous mitochondrial activity.

This naturally raises the questions of what confers on dFBNs the ability to respond flexibly to the varying metabolic states and how dFBNs can implement such a response through mitochondrial dynamics.

dFBNs have higher OxPhos gene expression levels and mitochondrial turnover than most other neuronal populations (notably, KCs and PNs) (Davie et al., 2018). OxPhos levels correlate with highly fused mitochondrial networks (Youle & Van Der Bliek, 2012), which support efficient energy production (Hoitzing et al., 2015; Picard et al., 2013), and by removing defective mitochondria, the mitochondrial turnover prevents the build-up of dysfunctional organelles that could hamper cellular energy production (Pickles et al., 2018).

The observation itself that sleep loss provokes mitophagy and fission in dFBNs, rather than stress-induced hyperfusion (Friedman & Nunnari, 2014; Gomes et al., 2011; Rambold et al., 2011; Tondera et al., 2009), which is the usual response to mild insults (Shutt & McBride, 2013), hints at extensive oxidative damage.

These observations suggest that dFBNs are tuned to high energy demand and are more sensitive to perturbation than other cell types, possibly sensing organism-wide metabolic states precociously. This enables them to dynamically adjust their activity accordingly, thus capturing the internal metabolic state cell-autonomously. It is plausible that this process is implemented by, among other mechanisms, harbouring mitochondria in elevated continuous turnover, offering multiple control points to generate a wide dynamic range of shapes and numbers suitable to promptly respond via changes in abundance and fission/fusion balance when required. In this sense, it is not surprising that dFBNs are able to retain 'metabolic memories', estimating time elapsed from last feeding (Flores-Valle et al., 2025) or, when challenged by time-restricted feeding (Dissel et al., 2022) or protein deprivation (Goldschmidt et al., 2023), multiplexing the representation of two intertwined internal states like the mammalian hypocretin/orexin neurons (Willie et al., 2001).

## Outlook

dFBNs of the fly brain are currently the only identified neurons where the balance between mitochondrial fission and fusion is bidirectionally coupled to sleep (Sarnataro et al., 2025). However, sleep history-related mitochondrial dynamics have been identified in other circuits, including the mammalian cortex (Aboufares El Alaoui et al., 2023; De Vivo et al., 2016). [Correction made on 29th September 2025, after first online publication: The phrase has been corrected to read "dFBNs of the fly brain"]

Investigating whether such changes are mere epiphenomena or rather instructive compensations would require manipulation of the mitochondrial dynamics machinery and monitoring of the corresponding behavioural readouts. For example, if the 'megamitochondria' observed in the sleep-deprived cortex, as much as the upregulation of mitochondrial and metabolic genes (Cirelli & Tononi, 1998; Nikonova et al., 2005, 2010), served to safeguard cells by reducing the accumulation of ROS (Wakabayashi, 2002) and delaying the apoptotic response (Yang et al., 2008), then artificially impeding or promoting their formation should lead to different homeostatic responses to sleep loss.

While this process has so far been identified in dFBNs, multiple brain areas, in the fly as well as in other species, have been pinpointed as being able to shape various aspects of sleep behaviour (architecture, homeostasis, intensity, duration, arousability, etc) and to be affected by sleep history.

If the various features of sleep are encoded in, or even just enacted by, several circuitries distributed across the brain with varying magnitudes and effect directions (Shafer & Keene, 2021), then the analyses of their sub-cellular substrates and mechanisms ought to be cell-type specific. The fruit fly model has a privileged position in this regard: the stereotyped specification of neurons creates neuronal networks with reliably targetable and identifiable signposts; the availability of whole-brain electron microscopy datasets for connectomics (with annotated mitochondrial outlines) (Dorkenwald et al., 2024; Scheffer et al., 2020) and single-cell transcriptomes (Croset et al., 2018; Dopp et al., 2024; Li et al., 2022) elucidates the wiring diagram of the brain network and the cellular identity of each node. Examining these brain-wide maps looking for gradients or correlations of cellular properties, for example mitochondrial shapes and anatomical/sub-cellular distribution, levels of OxPhos and post-translational modifiers of the core machinery of the mitochondrial dynamics, expression of antioxidant or mitochondrial lipid-processing enzymes, in vivo dynamics (Loring et al., 2020) of metabolites, possibly in brains belonging to animals experiencing various levels of sleep pressure, might pinpoint novel mechanisms and circuits relevant to this behaviour.

Interestingly, sleep is not the only behaviour whose main control circuitry displays mitochondrial dynamics with a covarying coupling to the behavioural state.

Hunger, another homeostatically controlled internal state with obvious metabolic repercussions, is reflected in the mitochondrial dynamics of two mammalian hypothalamic nuclei involved, with opposing roles, in the regulation of energy and glucose metabolism: the hunger-promoting agouti-related peptide and neuropeptide-Y (Agrp/NPY) neurons and hunger-suppressing pro-opiomelanocortin (POMC) neurons. Mitochondria in Agrp/NPY neurons fuse when the animals are overfed, while knockout of mitofusin-2 in these cells impairs their electrical activity and weight gain (Dietrich et al., 2013); POMC neurons display opposite rearrangements upon hampering of fusion: hyperphagia, obesity (Schneeberger et al., 2013), and reduced sensitivity for leptin and glucose (Santoro et al., 2017).

The creation of long-term memory (LTM) elicits mitochondrial pyruvate flux in the axonal compartment of KCs of the fly mushroom body (Plaçais et al., 2017) and relies on the relocation of some mitochondria out of the neuronal cell body and increased mitochondrial motility in the axonal compartment. Impairing mitochondrial dynamics abolishes the LTM-evoked increased pyruvate consumption and mitochondrial transport in the mushroom body axonal compartment, thus preventing LTM formation (Pavlowsky et al., 2024).

The circadian clock dictates the periodic fluctuations of tissues' functions, including mitochondrial morphology (Schmitt et al., 2018): in the hypothalamic suprachiasmatic nucleus (SCN), the central clock pacemaker of the mammalian brain, calcium levels fluctuate in a circadian fashion (Colwell, 2000; Ikeda et al., 2003) and deletion of mitofusin-2 in SCN impairs circadian periodicity and sleep (Stoiljkovic et al., 2025).

In these examples, neurons experience varying metabolic demands in different subcellular compartments, influenced by the brain's activity linked to a cognitive process. For behaviours with a direct metabolic outcome, deploying the mitochondrial morphofunction in its own neuronal substrate to influence its activity, as in the case of sleep and dFBNs, seems to be a particularly convenient feedback mechanism.

Thus, mitochondrial dynamics provide a pathway to flexibly respond to the varying metabolic demands driven by behaviour: they reshuffle metabolic fluxes within the constraining costs of neural information (Laughlin et al., 1998).

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

## Additional information

### Competing interests

No competing interests declared.

### Author contributions

Sole author.

### Funding

Wellcome Trust (WT): Raffaele Sarnataro, 215200/Z/19/Z.

### Acknowledgements

I would like to thank P. Hasenhuetl, R. Klemm, and G. Miesenböck for feedback on this manuscript, and The Physiological Society for funding the meeting which prompted this collection of reviews and for granting me a Conference Attendance Award supporting my participation.

### Keywords

ATP, energy, homeostasis, metabolism, mitochondria, neurobiology, neuron, sleep

## Supporting information

Additional supporting information can be found online in the Supporting Information section at the end of the HTML view of the article. Supporting information files available:

**Peer Review History**

