## [Peer Review History · The Journal of Physiology]

Neurobiology of mitochondrial dynamics in sleep

Raffaele Sarnataro

DOI: 10.1113/JP288054

Corresponding author(s): Raffaele Sarnataro (raffaele.sarnataro@cncb.ox.ac.uk)

Review Timeline:

Submission Date:	26-May-2025
Editorial Decision:	27-Jun-2025
Revision Received:	27-Jul-2025
Accepted:	04-Aug-2025

Senior Editor: Laura Bennet

Reviewing Editor: Diana Martinez

Transaction Report:

Dear Dr Sarnataro,

Re: JP-SR-2025-288054 "**Neurobiology of mitochondrial dynamics in sleep**" by Raffaele Sarnataro

Thank you for submitting your manuscript to The Journal of Physiology. It has been assessed by a Reviewing Editor and by 2 expert referees and we are pleased to tell you that it is acceptable for publication following satisfactory revision.

ABSTRACT FIGURES: Authors may use The Journal's premium BioRender account to create/redraw their Abstract Figures (and any other suitable schematic figures). Information on how to access this account is here: <https://physoc.onlinelibrary.wiley.com/journal/14697793/biorender-access>.

REVISION CHECKLIST: Upload a full Response to Referees file. To create your 'Response to Referees' copy all the reports, including any comments from the Senior and Reviewing Editors, into a Microsoft Word, or similar, file and respond to each point, using font or background colour to distinguish comments and responses and upload as the required file type.

- 'Potential Cover Art' for consideration as the issue's cover image.
- Appropriate Supporting Information (Video, audio or data set: see https://jp.msubmit.net/cgi-bin/main.plex?form_type=display_requirements#supp).

We look forward to receiving your revised submission.

Yours sincerely,

EDITOR COMMENTS

Reviewing Editor:

Comments to the Author:

Dr. Sarnataro has authored a very interesting and well-written review on mitochondrial dynamics in sleep. The manuscript was reviewed by two expert reviewers and the reviewing editor. The author integrates a multiple species approach to examining the modulation of mitochondrial fission, fusion, and mitophagy in the sleep-wake cycle. The figures are very clear and provide support for much of the review. The manuscript was easy to read and topics flowed very well. However, reviewers had some comments concerning some contradictory points and information within the text including the energetic demand of synaptic activity and lactate metabolism. There are also some grammatical issues that are easily addressed.

REFEREE COMMENTS

Referee #1:

The manuscript "Neurobiology of mitochondrial dynamics in sleep", authored by Raffaele Sarnataro, discusses the modulation of mitochondrial fission, fusion, and mitophagy in the context of neuronal energy demands (i.e., sleep-wake). The text is clear and well-structured, integrating scientific findings from various species, including humans, mice, and *Drosophila melanogaster*.

My main criticism concerns the statement that increased mitophagy occurs during neuronal synaptic activity, thereby inducing a metabolic shift toward glycolysis. However, as discussed earlier in the review, synaptic activity is a condition of extremely high energy demand, during which mitochondrial fusion is typically observed. Since fusion acts as a protective mechanism against mitophagy, this point appears contradictory and needs clarification.

Another important aspect is the lack of discussion regarding the role of lactate metabolism. Neurons are highly dependent on lactate derived from astrocytic metabolism. The absence of this topic represents a significant gap in the manuscript. A better contextualization of both glucose and lactate metabolism in relation to mitochondrial dynamics and sleep could greatly enhance our understanding of neuronal energy metabolism.

In addition, little attention is given to intracellular pathways and hormones involved in the regulation of mitochondrial dynamics during sleep. For instance, the manuscript mentions that "loss of sleep alters mitochondrial and metabolic gene expression levels in the brain (Cirelli & Tononi, 1998; Nikonova et al., 2005, 2010; Aboufares El Alaoui et al., 2023)", but does not specify which genes are affected. Citing specific genes related to energy metabolism and mitochondrial dynamics from these studies would strengthen the discussion.

Lastly, the meaning of the phrase "The elevated intracellular redox state"* in line 380 is unclear. Does it refer to a more oxidized or more reduced state? This should be clarified for better understanding.

Referee #2:

I found this article to be highly enjoyable and full of interesting observations from the literature and ideas about how mitochondrial dynamics may be related to sleep and its functions. The work is fairly comprehensive given its length, and bridges nicely data from both mammalian and fly models.

The work is quite readable, although in a few places perhaps only specialists will be able to fully appreciate the nuances of the discussion. Grammatically, there are a few areas that need improvement, but overall the quality is already high.

Specific notes/suggestions (these are mostly small grammatical issues):

line 100-101. Something is missing about the cortical neurons. The "for example" just lacks the follow-up, with dendrites mentioned immediately after

line 111-- "This" but it is not 100% clear what in the antecedent "this" refers to.

line 156-161. This section is very difficult to understand. Also, in general, I find the use of the colon (:) is not quite right, grammatically, throughout the text, and that may be part of the source of the difficulty to understand this point of this section.

line 170-173 This run-on sentence is difficult to parse.

line 192-195. This is another section that is hard to understand, due to grammatical issues.

198-205. These appear to be a series of disjointed microparagraphs, one sentence each. It is hard to follow the points, since they do not seem connected by a logical flow of a paragraph (thesis statement, evidence, conclusion, e.g.)

line 373-375. This is too vague. "it", "opposite effects". More clarity is needed here.

line 191-- "Specularly" appears to be a typo

line 394-400. This introduces some concepts that are not clearly explained. For example, ring neurons and recurrent circuitry therein will only be clear to experts in fly sleep circuits.

Overall, I find the figures very clear.

END OF COMMENTS

EDITOR COMMENTS

Reviewing Editor:

Comments to the Author:

Dr. Sarnataro has authored a very interesting and well-written review on mitochondrial dynamics in sleep. The manuscript was reviewed by two expert reviewers and the reviewing editor. The author integrates a multiple species approach to examining the modulation of mitochondrial fission, fusion, and mitophagy in the sleep-wake cycle. The figures are very clear and provide support for much of the review. The manuscript was easy to read and topics flowed very well. However, reviewers had some comments concerning some contradictory points and information within the text including the energetic demand of synaptic activity and lactate metabolism. There are also some grammatical issues that are easily addressed.

We thank the editor and the reviewers for their work.

We have addressed all the points raised by the reviewers regarding both some content clarifications or additions, and the flow of some sentences. Our responses are detailed point-by-point below.

REFEREE COMMENTS

Referee #1:

The manuscript "Neurobiology of mitochondrial dynamics in sleep", authored by Raffaele Sarnataro, discusses the modulation of mitochondrial fission, fusion, and mitophagy in the context of neuronal energy demands (i.e., sleep-wake). The text is clear and well-structured, integrating scientific findings from various species, including humans, mice, and *Drosophila melanogaster*.

Thank you.

My main criticism concerns the statement that increased mitophagy occurs during neuronal synaptic activity, thereby inducing a metabolic shift toward glycolysis. However, as discussed earlier in the review, synaptic activity is a condition of extremely high energy demand, during which mitochondrial fusion is typically observed. Since fusion acts as a protective mechanism against mitophagy, this point appears contradictory and needs clarification.

We thank the referee for highlighting this point. Mitophagy has been observed in condition of unphysiologically exacerbated activity, like excitotoxicity (Van Laar et al., 2015), artificially boosted OxPhos (Han et al., 2021), and seemingly under synaptic scaffolding impairment (Jang et al., 2016). For space constraints, simplicity of discourse, and since we focus on physiological processes, we have removed the reference that links mitophagy to elevated synaptic activity.

Another important aspect is the lack of discussion regarding the role of lactate metabolism. Neurons are highly dependent on lactate derived from astrocytic metabolism. The absence of this topic represents a significant gap in the manuscript. A better contextualization of both glucose and lactate metabolism in relation to mitochondrial dynamics and sleep could greatly enhance our understanding of neuronal energy metabolism.

Thanks for highlighting the importance of this topic that we had only mentioned for space reasons. We now refer prominently to the astrocyte-neuron lactate shuttle in the paragraph "Neuronal bioenergetics of sleep" and mention the involvement of lactate in promoting mitochondrial biogenesis in the following paragraph, in the context of mitochondrial dynamics in cortex.

In addition, little attention is given to intracellular pathways and hormones involved in the regulation of mitochondrial dynamics during sleep. For instance, the manuscript mentions that "loss of sleep alters mitochondrial and metabolic gene expression levels in the brain (Cirelli & Tononi, 1998; Nikonova et al., 2005, 2010; Aboufares El Alaoui et al., 2023)", but does not specify which genes are affected. Citing specific genes related to energy metabolism and mitochondrial dynamics from these studies would strengthen the discussion.

We now explicitly mention many of the genes and pathways differentially regulated by loss of sleep in the brain.

Lastly, the meaning of the phrase "The elevated intracellular redox state"* in line 380 is unclear. Does it refer to a more oxidized or more reduced state? This should be clarified for better understanding.

We have clarified it refers to a more oxidized state

Referee #2:

I found this article to be highly enjoyable and full of interesting observations from the literature and ideas about how mitochondrial dynamics may be related to sleep and its functions. The work is fairly comprehensive given its length, and bridges nicely data from both mammalian and fly models.

The work is quite readable, although in a few places perhaps only specialists will be able to fully appreciate the nuances of the discussion. Grammatically, there are a few areas that need improvement, but overall the quality is already high.

Thank you.

Specific notes/suggestions (these are mostly small grammatical issues):

line 100-101. Something is missing about the cortical neurons. The "for example" just lacks the follow-up, with dendrites mentioned immediately after

We have adjusted the focus of the paragraph to first highlight the common occurrence of various mitochondrial shapes through neuronal arborisations, and then move to discussing the functional consequences.

line 111-- "This" but it is not 100% clear what in the antecedent "this" refers to.

We have changed the sentence to make the references more explicit.

line 156-161. This section is very difficult to understand. Also, in general, I find the use of the colon (:) is not quite right, grammatically, throughout the text, and that may be part of the source of the difficulty to understand this point of this section.

We have split the sentence into shorter ones and changed the punctuation for clarity (we use colon elsewhere in the text for explanations, as per its prescribed use).

line 170-173 This run-on sentence is difficult to parse.

We have split the sentence into two shorter ones and changed the wording for clarity.

line 192-195. This is another section that is hard to understand, due to grammatical issues.

We have split the sentence into two shorter ones and changed the wording for clarity.

198-205. These appear to be a series of disjointed microparagraphs, one sentence each. It is hard to follow the points, since they do not seem connected by a logical flow of a paragraph (thesis statement, evidence, conclusion, e.g.)

We have split the sentence into shorter ones and changed the wording for clarity.

line 373-375. This is too vague. "it", "opposite effects". More clarity is needed here.

We have rendered both the sentence subject and the "opposite effects" explicit for clarity.

line 191-- "Specularly" appears to be a typo

We have changed it to "symmetrically"

line 394-400. This introduces some concepts that are not clearly explained. For example, ring neurons and recurrent circuitry therein will only be clear to experts in fly sleep circuits.

We have extended the paragraph to elaborate in more detail on the recurrent circuitry to improve clarity for non-experts in fly neuroscience.

Overall, I find the figures very clear.

Thank you.

References

Han S, Zhang M, Jeong YY, Margolis DJ & Cai Q (2021). The role of mitophagy in the regulation of mitochondrial energetic status in neurons. *Autophagy* 17, 4182–4201.

Jang S, Nelson JC, Bend EG, Rodríguez-Laureano L, Tueros FG, Cartagena L, Underwood K, Jorgensen EM & Colón-Ramos DA (2016). Glycolytic Enzymes Localize to Synapses under Energy Stress to Support Synaptic Function. *Neuron* 90, 278–291.

Van Laar VS, Roy N, Liu A, Rajprohat S, Arnold B, Dukes AA, Holbein CD & Berman SB (2015). Glutamate excitotoxicity in neurons triggers mitochondrial and endoplasmic reticulum accumulation of Parkin, and, in the presence of N-acetyl cysteine, mitophagy. *Neurobiology of Disease* 74, 180–193.

Dear Dr Sarnataro,

Re: JP-SR-2025-288054R1 "**Neurobiology of mitochondrial dynamics in sleep**" by Raffaele Sarnataro

I am pleased to tell you that your Symposium Review article has been accepted for publication in The Journal of Physiology, subject to any modifications to the text that may be required by the Journal Office to conform to House rules.

NEW POLICY: In order to improve the transparency of its peer review process, The Journal of Physiology publishes online as supporting information the peer review history of all articles accepted for publication. Readers will have access to decision letters, including all Editors' comments and referee reports, for each version of the manuscript and any author responses to peer review comments. Referees can decide whether or not they wish to be named on the peer review history document.

The last Word version of the paper submitted will be used by the Production Editors to prepare your proof. When this is ready, you will receive an email containing a link to Wiley's Online Proofing System. The proof should be checked and corrected as quickly as possible.

All queries at proof stage should be sent to tjp@wiley.com.

The accepted version of the manuscript is the version that will be published online until the copy edited and typeset version is available. Authors should note that it is too late at this point to offer corrections prior to proofing. Major corrections at proof stage, such as changes to figures, will be referred to the Reviewing Editor for approval before they can be incorporated. Only minor changes, such as to style and consistency, should be made a proof stage. Changes that need to be made after proof stage will usually require a formal correction notice.

Are you on Twitter? Once your paper is online, why not share your achievement with your followers. Please tag The Journal (@jphysiol) in any tweets and we will share your accepted paper with our 30,000+ followers!

If you would like to receive our 'Research Roundup', a monthly newsletter highlighting the cutting-edge research published in The Physiological Society's family of journals (The Journal of Physiology, Experimental Physiology and Physiological Reports), please click this link, fill in your name and email address and select 'Research Roundup':
<https://www.physoc.org/journals-and-media/membernews/>

Yours sincerely,

Laura Bennet
Senior Editor
The Journal of Physiology

EDITOR COMMENTS

Reviewing Editor:

Comments to the Author:

We thank the author for their resubmission. The reviewers and reviewing editor note that their concerns have been addressed. The manuscript is well-written and synthesizes recent studies and information concerning mitochondria and sleep.

REFEREE COMMENTS

Referee #1:

The authors answered my questions and accepted my suggestions.

Referee #2:

All the changes I recommended have been made and the paper is even stronger now.

* IMPORTANT NOTICE ABOUT OPEN ACCESS *

To assist authors whose funding agencies mandate public access to published research findings sooner than 12 months after publication, The Journal of Physiology allows authors to pay an open access (OA) fee to have their papers made freely available immediately on publication.

You will receive an email from Wiley with details on how to register or log-in to Wiley Authors Services where you will be able to place an OnlineOpen order.

You can check if your funder or institution has a Wiley Open Access Account here: <https://authorservices.wiley.com/author-resources/Journal-Authors/licensing-and-open-access/open-access/author-compliance-tool.html>.

Your article will be made Open Access upon publication, or as soon as payment is received.

If you wish to put your paper on an OA website such as PMC or UKPMC or your institutional repository within 12 months of publication you must pay the open access fee, which covers the cost of publication.

OnlineOpen articles are deposited in PubMed Central (PMC) and PMC mirror sites. Authors of OnlineOpen articles are permitted to post the final, published PDF of their article on a website, institutional repository, or other free public server, immediately on publication.

Note to NIH-funded authors: The Journal of Physiology is published on PMC 12 months after publication, NIH-funded authors DO NOT NEED to pay to publish and DO NOT NEED to post their accepted papers on PMC.